# Variation in Assessment of Leaf Pigment Content from Vegetation Indices Caused by Positions and Widths of Spectral Channels

**DOI:** 10.3390/plants14213355

**Published:** 2025-10-31

**Authors:** Alexander Machikhin, Anastasia Zolotukhina, Georgiy Nesterov, Daria Zdarova, Anastasia Guryleva, Oksana Gusarova, Sergei Ladan, Vladislav Batshev

**Affiliations:** Scientific and Technological Center of Unique Instrumentation, Russian Academy of Sciences, 15 Butlerova, 117342 Moscow, Russia; machikhin@ntcup.ru (A.M.); nesterov.gv@ntcup.ru (G.N.); zdarova.da@ntcup.ru (D.Z.); guryleva.av@ntcup.ru (A.G.); gusarova.oa@ntcup.ru (O.G.); s.ladan@bk.ru (S.L.); batshev_vlad@mail.ru (V.B.)

**Keywords:** spectral reflectance, vegetation index, spectroscopy, data processing, chlorophylls, carotenoids

## Abstract

Vegetation indices (VIs) are a widely adopted and straightforward tool for non-contact estimation of chlorophyll and carotenoid content in plant leaves. However, VI-based method accuracy depends critically on instrument configuration and calibration procedures. This study aimed to evaluate the sensitivity of VI-based pigment assessment to variations in spectral channel parameters (central wavelength and bandwidth) as well as to changes in calibration details defined by the specific VI formula. Pigment content was measured in leaves of *Lactuca sativa* L. and *Cucumis sativus* L. at contrasting developmental stages using VI-based reflection spectroscopy across the 450–950 nm spectral range with various protocols and spectrophotometry as the reference method. VI values were calculated with varying central wavelength and widths of spectral bands, and across different VI formulas. Comparative analysis of the obtained measurements revealed that even minor shifts in central wavelengths of less than 20 nm or the use of an alternative index formula could lead to relative errors of 42–77% in the estimation of chlorophylls and carotenoids content, while changes in bandwidth had a much smaller impact, resulting in only 2–5% relative errors. Even with identical parameters of spectral channels, the choice of an appropriate VI and its regression model could introduce significant errors, ranging from 36% to 86%. These findings highlight the critical role of instrument specifications and calibration models in the VIs-based method accuracy and stability, as measurement errors can lead to suboptimal agronomic decisions. Moreover, our study underscores that comparing results from different sensors or platforms can be unreliable unless the channel parameters and calibration details are clearly specified. Therefore, standardization and transparency in VIs assignment is vital to ensure reproducibility and cross-compatibility in non-destructive pigment monitoring by using various devices.

## 1. Introduction

Pigment content in plant leaves is one of the key physiological parameters that is highly sensitive to changes in various metabolic and developmental processes of the plant organism [1]. Assessment of the chlorophylls (*Chl*) and carotenoids (*Car*) concentrations is a well-established procedure in precision agriculture, including application of agrochemicals [2], optimization of agronomic practices [3], yield prediction [4], and biofortification of crops [5], as well as early detection of biotic and abiotic stresses [6]. Conventional methods for pigment assessment, such as spectrophotometry and high-performance liquid chromatography, provide high accuracy but are destructive, time-consuming, laboratory-intensive and limited to laboratory conditions [7]. Therefore, there is a significant trend towards the development of ground-based and remote sensing approaches that allow high-throughput and non-destructive estimation of leaf pigment concentration directly in the field. Low-cost implementations of such systems are also in high demand for on-farm monitoring [8].

One of the key concepts in non-destructive quantitative assessment of pigment content is reflectance spectroscopy. The method relies on the correlation between the pigments content in the leaf and its spectral properties. For the data interpretation in this approach, the spectral reflection characteristic is described by a quantitative metric—VI (Vegetation Index) [9]. VI is calculated as a combination of reflectance values at several wavelengths exhibiting the highest sensitivity to the target parameter. The established workflow of VI-based reflection spectroscopy includes the main stages presented in Figure 1. The first stage is preliminary and is usually carried out during the device development or updating the methodology for new objects. It involves: obtaining ground truth data on pigments; recording spectral characteristics; determining the VI and subsequent calibration curve that provide the most accurate pigment calculation. Next, during routine measurements with a ready-made device, spectral data from arbitrary plants are collected, and the preferred index is calculated. The final stage involves calculation of the pigment concentration from the VI applying the calibration curve. Each stage of the procedure contains potential sources of error; however, this study aims to highlight the critical role of instrument specifications and calibration procedures in ensuring the accuracy and stability of VIs-based methods.

This concept has been implemented in instruments with various designs, depending on the objectives, scale and performance of pigment monitoring. Handheld contact chlorophyll analyzers such as SPAD and PAM [10] require inserting the leaf into a measurement module. Handheld devices for remote pigment assessment, such as GreenSeeker [11] acquire light reflected from the canopy, which is illuminated. These compact handheld sensors typically calculate the values of one or a few VIs based at red and near-infrared wavelengths and are primarily used in proximal sensing for diagnosing plants grown under controlled environments, mainly greenhouses or on small experimental plots. In climate chambers, systems with higher spectral and spatial resolution are also in demand, for instance, used in breeding studies, as they allow detection of subtle differences between genotypes. Hyperspectral imaging systems offer VI mapping and allow the estimation of pigment content at the pixel level [12]. Multispectral cameras record similar three-dimensional spatio-spectral data with lower spectral resolution but with higher performance due to the simultaneous acquisition of spectral images. These devices can be mounted on agricultural machinery [13] or unmanned aerial vehicles [14,15] and are particularly suited for monitoring large areas, covering thousands of hectares. Thus, VI-based approaches represent a versatile framework, ranging from handheld leaf-clip sensors to large-scale imaging systems. However, their accuracy remains dependent on calibrations results, device design and environmental conditions, in addition to non-pigment parameters [16].

To enhance the sensitivity to variations in pigment concentration, specialized VIs have been developed for estimating chlorophylls [17] and carotenoids [18] content. The most of existing indices are two-band ratios that combine spectral regions sensitive to pigment content (400–740 nm) with relatively insensitive areas (750–900 nm) [19,20]. Typical forms of such indices are presented in [21] and include the Simple Ratio (SR), Difference Index (DI), Modified Simple Ratio (mSR), Modified Normalized Difference Index (mNDI), Triangular Vegetation Index (TVI), Soil-Adjusted Vegetation Index (SAVI), and the Normalized Difference Vegetation Index (NDVI). However, when a VI is determined using a spectral instrument, its value depends not only on the reflectance spectrum but also on the instrument’s channels, such as the central wavelength λ of the transmission function and its bandwidth (full width at half maximum) Δλ (Figure 2).

Pigment measurements are performed by many various devices that may differ in the number of spectral channels, their positions, and bandwidths, resulting in instrument-specific VI values. Table 1 presents the parameters of some VI-based tools and their parameters. The same index (e.g., NDVI) can be calculated from different “red” or “NIR” channels, which in practice cover various portions of the spectrum. Instrument calibration—which includes preprocessing and the selection of the optimal VI equation—also significantly affects the results, as demonstrated in [22]. These factors raise concerns about the validity of direct comparison of the measurements obtained by different instruments and highlight the variable performance of VI-based methods across devices. In addition, several studies [23] have shown that the effectiveness of a given VI can vary across plant species, due to morphological leaf traits, such as venation patterns, surface structure, pubescence, wax layers, leaf thickness, and water content. Ignoring these instrument- and species-specific differences can lead to misinterpretation: plant features and status may be over- or underestimated. Such errors may transform into non-optimal agronomic decisions, including excessive or insufficient fertilizer application, inappropriate irrigation timing or dosage, and ineffective crop protection treatments. This may not only reduce the efficiency of agricultural practices and increase costs but may also induce plant stress, accelerate soil depletion, and cause excessive accumulation of agrochemicals in the ecosystem.

Research on the optimization of VI-based approaches covers several key directions: designing new indices with specific formulas and wavelengths [18,31]; refining spectral channel combinations in existing formulas [32,33]; evaluating established vegetation indices for novel applications [2,34]; and determining the optimal calibration curve, which may be performed separately or in conjunction with the aforementioned points [17,35]. The principal focus of such research is to identify parameters that ensure robust predictive performance for specific tasks. The sensitivity of VI-based approaches to alterations in spectral channel parameters is typically presented as supplementary information and lacks systematic analysis [36,37].

Despite this, numerous studies highlight the importance of accounting for instrumental components and calibration in measurement results [38]. A quantitative analysis of the impact of several device characteristics, such as bandwidth, central wavelength, spatial resolution, and sensor degradation, is the primary focus of some works [39,40,41]. The majority of studies are devoted to assessing the sensitivity of the VI-based approach to the channel central wavelength for specific VI formulas [40,42], while the impact of bandwidth, or the combined effect of both parameters, is evaluated less frequently [43,44,45]. Most assessments are performed using simulated data [43,46]; some studies include experimental imaging with laboratory analysis [47] or integrate both simulated and experimental datasets [44,48]. Research on calibration optimization, that is, the selection of a vegetation index and calibration curve, is mainly aimed at identifying the combination with the highest predictive performance; however, some studies provide valuable, detailed information on the errors associated with all the combinations considered [5,49].

In the vast majority of such studies, VI-based approaches have been shown to be highly sensitive to each of the aforementioned measurement device characteristics [40,44,45,46]. Despite the presence of general trends, the magnitude of the identified errors and the recommendations regarding optimal characteristics vary among studies. Moreover, only a limited number of works consider multiple parameters simultaneously, even among those employing modeling [40,44]. Multi-crop studies, as well as ones examining variations in results across different developmental stages or target pigments, are rare [47,50]. The study design typically differs to such an extent that combining and generalizing their findings is challenging, as they vary in crop types, developmental stages, pigments, imaging scale, spectral resolution of measurement devices, and other factors. At the same time, systematic analysis and comparison of the relative contribution of the stability of individual device characteristics and the calibration to the accuracy of the VI-based approach should be conducted using data obtained under consistent conditions. This underscores the necessity of a comprehensive analysis of how channel characteristics and calibration influence VI-based pigment evaluation results derived from experimental data across different plant species, developmental stages, and pigments.

To address the issue mentioned above, the primary goals of this study were to: (1) obtain hyperspectral data and concentrations of different pigments for several crop species at various developmental stages; (2) develop regression models of pigments concentration as a function *f*(VI) to evaluate each combination of wavelengths involved in typical indices; (3) identify the optimal configuration of spectral channels, including their central wavelength and bandwidth; (4) assess the stability of pigments content measurements for the same channel parameters under different index formulas; and (5) assess the stability of pigments content measurements for the same Vis under different channel parameters. We believe our findings can inform sensor standardization and the design of cost-effective devices for breeders and farmers.

## 2. Results

The experimental dataset includes leaves of *Lactuca sativa* L. and *Cucumis sativus* L., representing differences both in variety and developmental stage. For each sample non-destructive spectral measurements of the leaf surface were conducted along with chemical analyses to quantify the pigment composition, including chlorophylls and carotenoids. This hybrid approach provided a reliable method for data collection and enabled thorough evaluation of the accuracy of VI–based pigment estimation.

### 2.1. Pigment Content

The average concentrations of chlorophylls and carotenoids for both species are summarized in Table 2. Presented data indicate that *Cucumis sativus* L. leaves generally have higher pigment concentrations. Standard deviations (approximately 30–40% of the mean values) reveal considerable variability within each group. In *Cucumis sativus* L., this variability indicates differences between developmental stages (BBCH 10–19 vs. BBCH 50–59), with young plants showing greater heterogeneity in pigment content compared to plants at the onset of flowering. In *Lactuca sativa* L., by contrast, most of the variability arises from cultivar differences: one cultivar accumulated markedly higher pigment levels, whereas the other showed lower values, which influenced the averaged results.

### 2.2. Reflectance Data

The application of spectral analysis to plant samples relies on the sensitivity of their reflectance spectra to variations in pigment concentration within leaves. This is shown through the comparison of *Cucumis sativus L*. and *Lactuca sativa* L. (Figure 3). The figure highlights the key absorption features of chlorophyll a, chlorophyll b, and β-carotene [51], as these pigments are primarily responsible for capturing light energy during photosynthesis and define the characteristic absorption peaks in the visible spectrum. Around 550 nm exhibits a more pronounced reflectance peak than around 700 nm, where the difference in red-edge shift between their spectral reflectance is noticeable. This spectral band corresponds to a minimum in chlorophyll absorption [52]; therefore, the sharper peak in *L. sativa* leaves reflects their lower overall pigment concentration compared with *C. sativus.* Such differences highlight the diagnostic potential of reflectance features near pigment absorption minima and maxima, providing a basis for developing vegetation indices that enhance the accuracy and robustness of pigment estimation.

### 2.3. Performance of a VI-Based Model at Various Positions and Bandwidths

We tested all combinations of spectral channel parameters, analyzing about 4,000,000 setups (roughly 250,000 per bandwidth). The wavelength range was 450–950 nm (step size: 1 nm). Bandwidths of the transmission functions were set to 1, 5, 10, 30, or 50 nm. As in [53], we used the coefficient of determination (*R*^2^) as our main optimization measure. *RMSE* was an additional check for accuracy and reliability. Switching from maximizing *R*^2^ to minimizing *RMSE* changed the best wavelengths by no more than 1 nm, and bandwidths remained unchanged, indicating our optimization was robust. A few exceptions were observed for certain indices: the DI for carotenoids in *Cucumis sativus* L. (optimal parameters: $\lambda_{1}$ = 707 nm, $\lambda_{2}$ = 728 nm, $\Delta \lambda_{1}$ = 50 nm, $\Delta \lambda_{2}$ = 1 nm), the mNDI ($\lambda_{1}$ = 451 nm, $\lambda_{2}$ = 666 nm, $\Delta \lambda_{1}$ = 50 nm, $\Delta \lambda_{2}$ = 1 nm), and mSR ($\lambda_{1}$ = 655 nm, $\lambda_{2}$ = 695 nm, $\Delta \lambda_{1}$ = 1 nm, $\Delta \lambda_{2}$ = 1 nm) indices for chlorophyll in *Cucumis sativus* L. However, in these three cases the *RMSE* difference for the models based on maximizing *R*^2^ to minimizing *RMSE* did not exceed 0.1 mg/L, and the *R*^2^ difference did not exceed 0.01. Since these exceptions account for less than 11% of the 28 optimization analyses performed, and given the small differences in the evaluation metrics, their influence on the study’s conclusions remained negligible.

The optimal wavelength combinations were then selected by maximizing *R*^2^ for estimating chlorophyll and carotenoid contents for each crop separately (Table 3 and Table 4). As pigments strongly correlate, informative wavelengths for both target parameters lie close to each other within a given plant but differ across species. We selected informative wavelengths at a bandwidth of 1 nm to capture the most contrasting spectral bands while minimizing interference from adjacent reflectance values. The observed differences in optimal model parameters for the two crops may result from variations in biophysical and biochemical leaf characteristics, such as cuticle properties, internal leaf structure, hydration status, leaf thickness, and other factors [10].

The optimal spectral channel parameters obtained in this study (central wavelength λ and bandwidth Δλ) for some formulas (SAVI, RI, mNDI) were compared with those reported in previous studies. Since several indices existed for a single formula, we selected the one whose λ and Δλ was closest to ours. However, little correspondence was found, mainly because the indices were originally developed for other agricultural crops (Table 5).

As shown in the tables, the optimal index type differed between crops, and the central wavelengths for the same index shifted depending on both crop and pigment. These results emphasize the need to analyze indices and wavelengths for each pigment–species combination to determine parameters that best fit a specific task. For each optimal index, we further examined the variability of prediction accuracy when applying alternative wavelengths (Figure 4). This approach allowed us to assess the robustness of indices to spectral parameter shifts and to identify the wavelength ranges in which their predictive performance remained stable.

As shown in Figure 4, even small shifts in the selected central wavelengths can lead to significant changes in prediction accuracy. A displacement of the central wavelength by less than 20 nm may reduce *R*^2^ from 0.8 to 0.3, indicating the loss of a meaningful relationship between the index and the pigment content. Such wavelength shifts also affect the accuracy of the predicted values, with deviations exceeding 50% of the actual concentration. Moreover, index variability differs between species, which emphasizes the need to account for species-related characteristics. In addition to wavelength position, the bandwidth ∆λ also influences significantly. To demonstrate this effect, the same index was evaluated at varying bandwidths (Figure 5). To prevent accuracy loss in practical applications, a comprehensive approach to fixation and monitoring of the recording device parameters is required. This involves maintaining the stability of the spectral curve of the active light source luminance and of the tunable filter transmittance during measurements, where applicable, as well as utilizing reference targets.

Figure 5 illustrates that the increase in the spectral channel width reduces the sensitivity of the index, while simultaneously improving its stability. This effect arises because, at narrow bandwidths, even a slight shift in the central wavelength of only a few nanometers can significantly reduce the predictive performance of the index. Therefore, the development of spectral indices requires careful consideration of both the central wavelength λ and the bandwidth ∆λ of the spectral channels. Moreover, the trade-off between bandwidth and stability should be mentioned during commercial sensor design.

### 2.4. Dependence of VI-Based Model Performance on the Formula of VI

The formula of the index itself is an essential factor, as it is incorporated during the calibration stage and directly affects the instrument’s metrological features. In this study, we first determined the optimal wavelengths used for calculating all considered indices. The model accuracy was then evaluated by varying only the index formula, without altering the wavelengths involved in the calculation (Figure 6).

Figure 6 demonstrates that the formula of a spectral index has a substantial impact on the results. *R*^2^ can vary from 0.9 to 0.02, leading to considerable errors in predicting pigment content. Numerous works typically report relatively high results, especially when employing advanced processing techniques such as convolutional neural networks [16,55]. This is likely because model training involves selecting optimal parameters for converting spectral characteristics into pigment concentrations, which shares certain similarities with the approach implemented in this study. Further studies should be devoted to examining the sensitivity of neural network approaches to spectral channel parameters. Figure 7 further illustrates this variability, showing the prediction results for 20 randomly selected samples from the datasets, each comprising 50 samples per group.

An incorrect choice of spectral index formula can result in prediction errors exceeding 100%. To address the instability of spectral indices, it is recommended to optimize both the index formula and the central wavelengths, taking into account the spectral bandwidth tailored to the specific pigment and crop under study.

## 3. Discussion

The obtained results substantiate the hypothesis that the VI-based approach to pigment estimation is sensitive to both the technical implementation of the instrument (central wavelengths and bandwidths of its spectral channels) and its calibration (VI formula and its relation to pigment content).

### 3.1. Effect of Spectral Channel Parameters

This study shows that even with a fixed VI formula and channel bandwidth, small shifts (20 nm) in central wavelengths can cause significant errors: *Chl* estimates may deviate by up to 77%, *Car*—up to 70% for *Cucumis sativus* L. (Figure 4a). For *Lactuca sativa* L., *Chl* estimates may deviate by up to 42%, *Car*—up to 68% (Figure 4b). The reason lies in the sharp variations in reflectance observed within pigment absorption regions; even minor wavelength shifts of less than 20 nm drastically impair the predictive capacity, leading to the loss of the VI–pigment relationship. Even small shifts (5 nm) from the optimal central wavelengths can slightly affect pigment estimates. For *Cucumis sativus* L., chlorophyll estimates may deviate by up to 5%, and carotenoid estimates by up to 1% For *Lactuca sativa* L., chlorophyll estimates may deviate by up to 4%, and carotenoids by up to 2%. When wavelength positions are fixed but bandwidths are varied, accuracy also declines (*Chl*—3%, *Car*—5% for *Cucumis sativus* L. and *Chl*—5%, *Car*—2% for *Lactuca sativa* L.). However, the impact of bandwidth ∆λ is generally less critical than the central wavelengths (Figure 5). These findings are consistent with [56], where PROSAIL simulations demonstrated that chlorophyll retrieval is highly sensitive to the configurations of visible and NIR channels. The differences in results between the two species are largely attributed to physiological characteristics. Leaf surface and internal structure, hydration status, and other factors influence the spectral response, leading to variations in errors [10]. However, the severity of deviations depending on spectral channel parameters remains consistent across both crops.

### 3.2. Effect of Calibration

The choice of VI formula and its regression model is even more influential. Even with identical spectral channels, *Chl* relative errors can reach 78%, *Car*—77% for *Cucumis sativus* L., and *Chl*—86%, *Car*—36% for *Lactuca sativa* L., depending on index formula (Figure 6). Because this study included plants with diverse morphologies and developmental stages, the results can be generalized across species. This agrees with previous reports [47,49], which showed that regression curves for pigment estimation vary widely among plant functional types and phenological stages. Thus, differences in cultivar characteristics and developmental stages lead to disparate regression curves, which makes universal calibrations unfeasible and calls for species- and stage-specific approaches.

### 3.3. Applicability of the VI-Based Concept

Despite these constraints, the VI-based concept has clear advantages. It is simple, fast, and well-suited for replacing hyperspectral systems with multispectral ones [48,57]. It can (a) significantly reduce data acquisition time and enable near-real-time measurements, (b) minimize redundancy and collinearity among channels, (c) lower equipment costs and support wider adoption in precision agriculture. However, these advantages are offset by fundamental limitations. The primary challenge is the ambiguity of VI values. Even when the same formula is applied, different instruments can deliver inconsistent values because of differences in spectral channels positions and bandwidths. As a result, identical plant samples may be assigned different pigment concentrations depending on the device used for analysis. This lack of standardization makes it difficult to unify VI-based methods across various platforms.

However, overcoming the mentioned limitations is essential, as the VI concept could serve as a basis for commercial device development. With a fixed and relatively limited number of required spectral channels, combined with minimal computational requirements when regression curves are predetermined, cost-efficient device implementation becomes feasible. Proper selection of design parameters and calibration procedures can ensure the reliability of such systems. Furthermore, careful attention should be paid to data distortion caused by variations in parameters not under investigation. For instance, leaf water status influences reflectance and may confound pigment estimation [58]. As shown earlier [16], selecting an appropriate vegetation index or integrating convolutional neural network models to estimate moisture content can improve robustness and expand the application to stress diagnostics.

### 3.4. Standardization

Several strategies could help to overcome these limitations, relevant not only for VI-based approaches but also for more advanced methods such as PLSR and machine-learning models [50,55,59]:(1)Index conversion frameworks. Developing methods to translate one VI into another would enable the comparison of results from different platforms. This requires reference spectral libraries simulating leaf reflectance at known pigments concentrations across multiple species and plant developmental stages.(2)Calibration standards. International standards are necessary to harmonize calibration procedures and reference targets, ensuring consistent results across instruments.(3)Instrument transparency and certification. Manufacturers should disclose spectral channel parameters, embedded VI formulas, and regression models. This would require a certification process to guarantee that pigment estimates are reliable and comparable across devices.

## 4. Materials and Methods

### 4.1. Plants

To investigate the relationship between leaf spectral characteristics and pigment concentration, several cultivars of lettuce (*Lactuca sativa* L.) were selected. They were different in leaf color and morphology. The early-maturing cultivar Credo (Oakleaf type) forms green, oak-shaped leaves with wavy, lobed edges, and the mid-early cultivar Moskovskiy Parnikoviy produces light-green leaves with a yellowish tint, blistered surface, and smooth margins. Plants were grown in an ebb-and-flow hydroponic system under a 14-h photoperiod with artificial illumination of 50 W. To ensure uniform leaf development and avoid morphological variation associated with early physiological disturbances, well-developed 30-day-old plants were used in the experiment. At this stage, the plants were divided into a control group and a treatment group, which were exposed to a nutrient solution with increased concentrations of calcium and magnesium salts. The baseline (control) solution contained macronutrients (N-9.72; P-0.82; K-3.97; Mg-1.19; S-1.22; Ca-2.88 mmol L^−1^) and trace elements (Fe-0.03; Zn-0.003; Cu-0.003; Mn-0.046; Mo-0.0006; Cl-0.29; B-0.02 mmol L^−1^). The EC of the nutrient solution was maintained at 1.2 mS cm^−1^ for lettuce. After a 5-day adaptation period, comparative measurements were carried out.

Plant spectral response may vary depending on the developmental stage. To address this, experiments were also performed with cucumber (*Cucumis sativus* L.) using the parthenocarpic early hybrid Mamlyuk F1 at different growth stages. Young plants at the BBCH 10–19 stage (development of the 4th–5th true leaf) and mature plants at BBCH 50–59 (onset of flowering) were selected for the experiment. The baseline (control) solution contained macronutrients (N-9.11; P-1.41; K-4.58; Mg-1.50; S-1.97; Ca-2.91 mmol L^−1^) and trace elements (Fe-0.05; Zn-0.006; Cu-0.006; Mn-0.077; Mo-0.001; Cl-0.18; B-0.035 mmol L^−1^). The electrical conductivity of the *Cucumis sativus* L. nutrient solution was maintained at 2.0 mS cm^−1^, and the pH of the solution was adjusted to 5.6 for both plant species. The description of the plants used in the experiment is provided in Table 6.

For spectral registration and subsequent imaging, ten plants were selected from each of the four experimental groups. From each plant, five fully developed leaves were excised, resulting in a total of fifty leaves per group. The leaves were uniform in coloration and free from local defects such as chlorosis, necrosis, or mechanical damage. All samples were taken from the upper, sun-exposed part of the canopy to ensure comparable light adaptation and minimize within-group variability. This standardized selection protocol was applied consistently across all groups.

To maintain consistent measurement geometry and prevent leaf wilting, leaves were excised and immediately placed in small containers filled with water, following the procedure described in [60]. After a 15-min equilibration period, spectral measurements were performed.

### 4.2. Chlorophyll and Carotenoid Analysis

The conventional approach for quantifying pigment content involves spectrophotometric analysis of an extract obtained through sample preparation. A standardized sample of leaf tissue (170 mg) is weighed, finely ground in a mortar, and then extracted with ethanol (25 mL). Measuring the optical density of the extract with a spectrophotometer allows the determination of total chlorophyll and carotenoid concentrations (mg/L) based on empirical relationships [61]:(1)$$Chl=6.1\times D_{665}+20.04\times D_{649},$$(2)$$Car=4.695\times D_{440.5}-0.268\times Chl,$$
where $D_{440.5}$, $D_{649}$ and $D_{665}$ represent the optical densities at 440.5 nm, 649 nm, 665 nm, respectively.

### 4.3. Spectral Data Acquisition

Spectral reflectance of leaves in 450–950 nm range was measured by Ocean Optics FLAME-VIS spectrometer. During measurements, both the leaf samples and a reference panel with uniform reflectance were illuminated by a halogen light source. The angle between the incident light and the measurement plane was set to 45° (Figure 8). The spectral signature of each leaf was obtained by averaging measurements from three areas of the leaf, carefully selected to avoid veins within the spectrometer’s 5 × 5° field of view.

All spectral measurements were carried out under controlled laboratory conditions with constant temperature and relative humidity. To minimize variability, measurements were taken at the same time of day (between 10:00 and 12:00 a.m.) and in the absence of external light sources or stray radiation. Recording the spectral reflectance of each leaf took approximately five minutes, including sample positioning, reference panel calibration, and data acquisition from three selected areas.

Three areas on each leaf were visually selected for uniform color and absence of local defects. Measurement spots were located between the veins at roughly equal distances from them, avoiding prominent veins, and generally positioned in the central lamina. To ensure a consistent orientation of the leaf relative to the light source and the spectrometer probe, a pressing glass was used to hold the leaf in place during measurements.

### 4.4. Data Processing

Figure 9 illustrates the main steps in processing the acquired spectral data. The leaf reflectance spectrum R(λ) was obtained by dividing the spectral radiance of the leaf $I_{obj}$(λ) by that of the reference panel $I_{ref}$(λ) followed by the removal of high-frequency noise using a Gaussian filter (σ = 20 nm).

Next, VIs were calculated using seven typical formulas reported in [21] for each combination of wavelengths $\lambda_{i}\;and\;\lambda_{j}$ within 450–950 nm range, with spectral channel widths of Δλ= 1 nm, 5 nm, 10 nm, 30 nm and 50 nm. The specified spectral range corresponds to that of modern spectral instruments [29,62]. Each spectral channel was modeled as a Gaussian function characterized by a specific central wavelength λ and bandwidth ∆λ. This approach allowed simulation of the effective spectral response of different sensors and accounted for the influence of both central wavelengths and bandwidths on the calculated VIs. To analyze the relationship between VIs and pigment content, a non-linear regression was employed. Specifically, an exponential model was assumed, reflecting the empirically observed saturation of many indices at high pigment concentrations [63].

Computation of a single model for one vegetation index formula and one wavelength combination required approximately 17 us, while generating models for all wavelength pairs for a single index (i.e., producing one *R*^2^, *RMSE*, and *RE* map) took about 0.9 s. All computations were performed on a computer equipped with an Intel Core Ultra 5 125H processor and 32 GB of LPDDR5X RAM, demonstrating the feasibility of the approach for high-throughput spectral analysis.

### 4.5. Model Evaluation

Statistical parameters used to evaluate the predictive performance of the model were calculated based on the predicted $\hat{y_{i}}$ and measured $y_{i}$ pigment contents as follows:(3)$$R^{2}=\frac{\sum_{i=1}^{n}\left(\hat{y_{i}}-\bar{y}\right)}{\sum_{i=1}^{n}\left(y_{i}-\bar{y}\right)}$$(4)$$RE=\frac{\left|\hat{y_{i}}-y_{i}\right|\times 100\%}{\bar{y}}$$(5)$$RMSE=\sqrt{\frac{{\sum_{i=1}^{n}\left(\hat{y_{i}}-\bar{y}\right)}^{2}}{n}}$$
where n denotes the number of measurements. The criterion for selecting the optimal model was the highest coefficient of determination *R*^2^ in combination with the lowest relative prediction error.

## 5. Conclusions

In this study, we assessed the sensitivity of pigment quantification using a VI-based approach to two key factors: (1) configuration of the spectral channels, including their central wavelength and bandwidth, and (2) device calibration procedure, specifically VI formula. Using *Cucumis sativus* L. and *Lactuca sativa* L. leaves as test samples, we conducted a comparative study of pigment content measurements accuracy. The comparison involved VI-based reflection spectroscopy under various protocols and spectrophotometry as the reference method. The results revealed that variations in instrumental or methodological parameters in the VI-based concept can result in substantial errors in pigment calculation ranging from 2% to 77%.

These findings emphasize the critical importance of complete transparency in instrument characteristics and embedded calibration models when employing VI-based methods. Data derived from different sensors or based on different calibration schemes cannot be reliably compared without this crucial information. Establishing common standards for channel configuration, calibration protocols, and index definitions will therefore be essential for improving the accuracy, reproducibility, and compatibility of optical tools for non-destructive pigment monitoring in plant science and precision agriculture. Moreover, the obtained outcomes could be used to develop cost-efficient multispectral sensors and calibration protocols for reliable pigment estimation across species and developmental stages.

## Figures and Tables

**Figure 1 plants-14-03355-f001:**
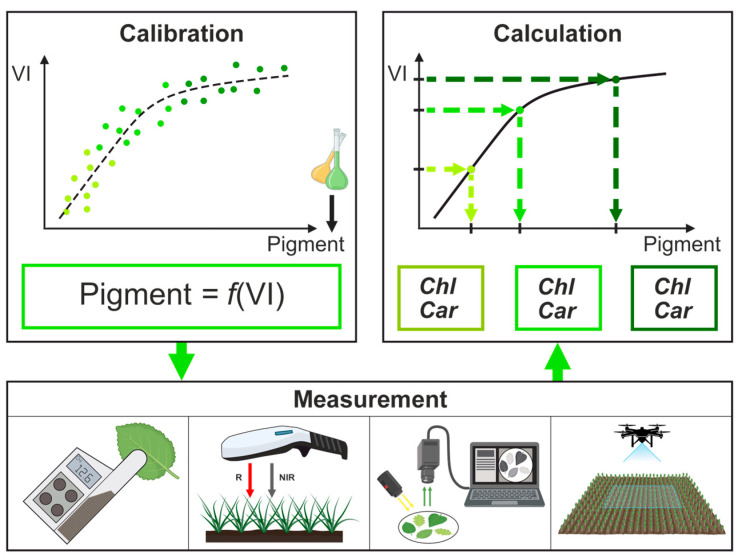
VI-based concept of pigment content measurement.

**Figure 2 plants-14-03355-f002:**
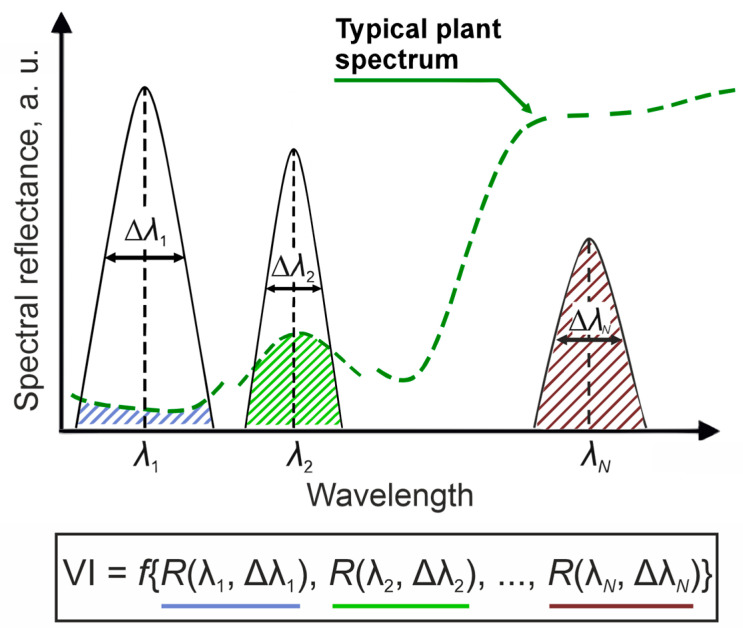
Principle of VI calculation.

**Figure 3 plants-14-03355-f003:**
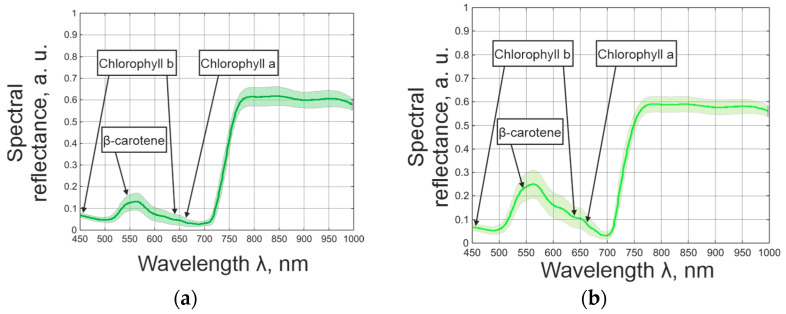
Spectral reflectance of (**a**) *Cucumis sativus* L. and (**b**) *Lactuca sativa* L. leaves.

**Figure 4 plants-14-03355-f004:**
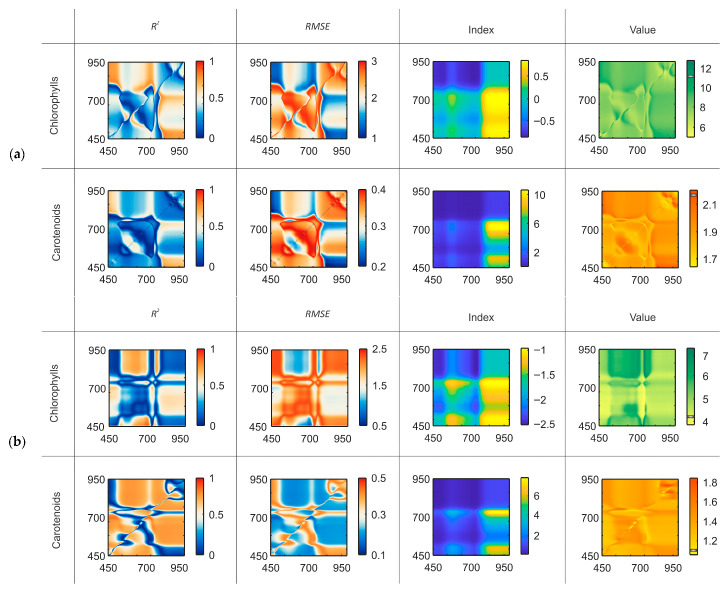
Maps after wavelength alteration showing index values (10 nm bandwidth), coefficients of determination (*R*^2^) and root mean squared error (*RMSE*), and corresponding index and pigment content maps for *Cucumis sativus* L. (**a**) and *Lactuca sativa* L. (**b**). The marker on the color bar of the pigment content map indicates the true concentration in the sample.

**Figure 5 plants-14-03355-f005:**
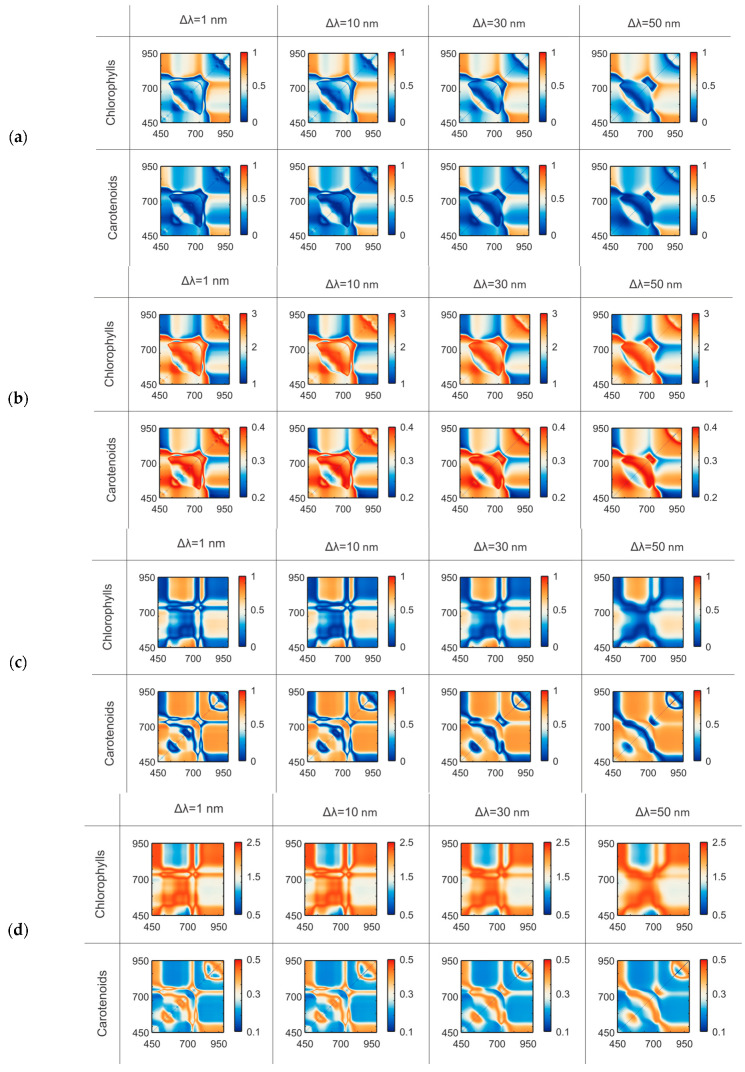
Dependencies of *R*^2^ and *RMSE* on channel widths ∆λ in *Cucumis sativus* L. (**a**,**b**) and *Lactuca sativa* L. (**c**,**d**).

**Figure 6 plants-14-03355-f006:**
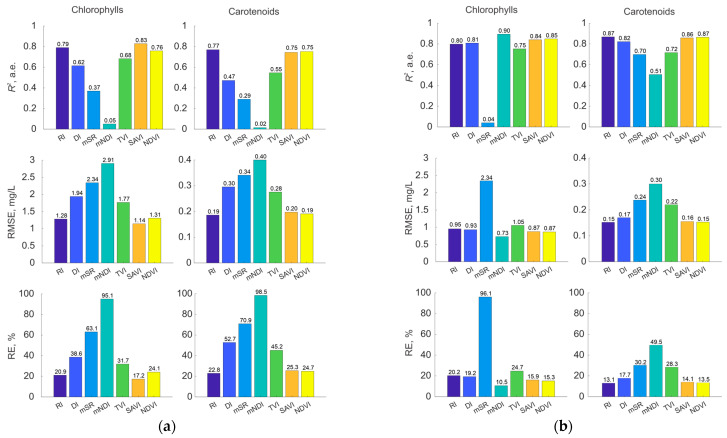
Comparison of *R*^2^, *RMSE* and *RE* for models using the optimal spectral index versus alternative forms for *Cucumis sativus* L. (**a**) and *Lactuca sativa* L. (**b**) at the same wavelengths.

**Figure 7 plants-14-03355-f007:**
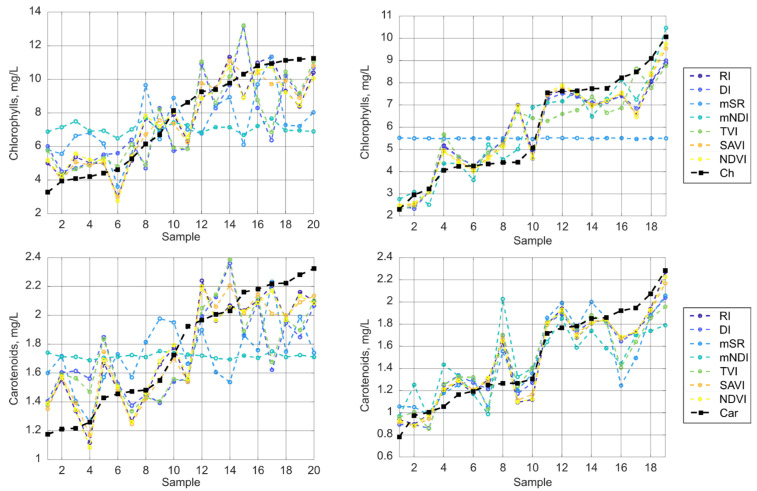
Prediction errors across different samples for various index formulas for *Cucumis sativus* L. (left side) and *Lactuca sativa* L. (right side) at the same combination of wavelengths.

**Figure 8 plants-14-03355-f008:**
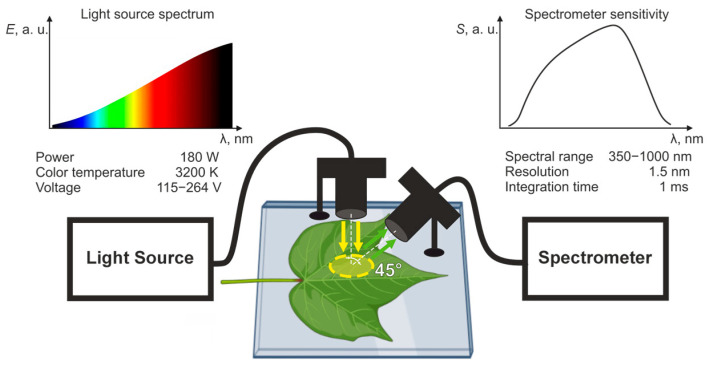
Experimental setup.

**Figure 9 plants-14-03355-f009:**
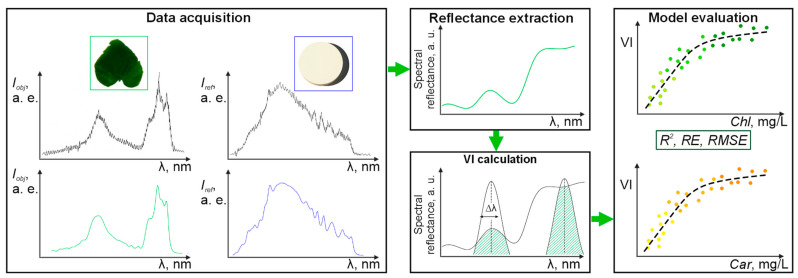
Data processing pipeline.

**Table 1 plants-14-03355-t001:** Some instrument for pigment estimation.

Instrument	Model, Manufacturer, Country	Central Wavelength/Bandwidth, nm	VI	Reference
Handheld Contact Sensor	SPAD-502Plus, Konica Minolta, Tokyo, Japan	650/30 940/10	DI	[24]
	Dualex 4, Force-A, Orsay, France	710/unknown 850/unknown	SR	[25]
Handheld Remote Sensor	PSR-1100F, Spectral Evolution, Haverhill, MA, USA	531/3 570/3	NDVI	[26]
	FieldSpec 3, ASD Inc., Falls Church, VA, USA	705/3 750/3	NDVI	[27]
Multispectral Imaging Sensor	Parrot Sequoia, Parrot, Paris, France	660/40 790/40	NDVI	[28]
	Micasense Altum, MicaSense Inc., Seattle, WA, USA	668/14 717/12	NDVI	[2]
Hyperspectral Imaging Sensor	Micro-Hyperspec VNIR, Headwall Photonics, Acton, MA, USA	515/6.4 550/6.4	SR	[29]
	Senop HSC-2, Senop Ltd., Kangsala, Finland	531/6 570/6	NDVI	[30]

**Table 2 plants-14-03355-t002:** Pigment values obtained via spectrophotometric measurement.

Crop	Maturity Stage	Total Number of Leaf Samples	Reference *Chl*, mg/L	Reference *Car*, mg/L
Lettuce (*Lactuca sativa* L.)/Credo	30 days juvenile	50	3.94 ± 0.83	1.13 ± 0.17
Lettuce (*Lactuca sativa* L.)/Moskovskiy Parnikoviy	30 days juvenile	50	8.24 ± 0.85	1.92 ± 0.18
Cucumber (*Cucumis sativus*)/Mamlyuk F1	BBCH 10–19 4–5 true leaves	50	7.88 ± 2.77	1.86 ± 0.31
Cucumber (*Cucumis sativus*)/Mamlyuk F1	BBCH 50–59 onset of flowering	50	7.53 ± 3.14	1.70 ± 0.46

**Table 3 plants-14-03355-t003:** Parameters of the optimal models for *Cucumis sativus* L. The index providing the highest *R^2^* is highlighted in bold.

Pigments	Formula	$\lambda_{1},\Delta \lambda_{1}$	$\lambda_{2},\Delta \lambda_{2}$	*R*^2^, a.e.	*RMSE*, mg/L	*RE*, %
*Chl*	SR	454, 1	752, 1	0.81	1.20	19
	DI	703, 10	725, 1	0.77	1.39	23
	mSR	751, 1	851, 1	0.61	1.98	39
	mNDI	713, 50	692, 1	0.79	1.39	21
	TVI	691, 50	715, 50	0.82	1.20	18
	**SAVI**	**478, 50**	**747, 1**	**0.83**	**1.14**	**17**
	NDVI	752, 1	453, 1	0.79	1.22	21
*Car*	**SR**	**451, 1**	**748, 1**	**0.77**	**0.19**	**23**
	DI	732, 1	526, 1	0.57	0.27	43
	mSR	451, 1	699, 1	0.54	0.27	46
	mNDI	716, 50	644, 1	0.63	0.24	37
	TVI	690, 50	714, 50	0.65	0.23	35
	SAVI	746, 1	456, 1	0.75	0.20	25
	NDVI	747, 1	451, 1	0.75	0.19	25

**Table 4 plants-14-03355-t004:** Parameters of the optimal models for *Lactuca sativa* L. The index providing the highest *R*^2^ is highlighted in bold.

Pigments	Formula	$\lambda_{1},\Delta \lambda_{1}$	$\lambda_{2},\Delta \lambda_{2}$	*R*^2^, a.e.	*RMSE*, mg/L	*RE*, %
*Chl*	SR	650, 1	672, 50	0.89	0.78	11
	DI	675, 50	649, 1	0.85	0.85	15
	mSR	601, 1	696, 1	0.74	1.24	26
	**mNDI**	**451, 1**	**649, 1**	**0.89**	**0.73**	**11**
	TVI	597, 1	771, 1	0.84	0.93	16
	SAVI	672, 50	650, 1	0.88	0.79	11
	NDVI	672, 50	650, 1	0.88	0.77	11
*Car*	**SR**	**650, 1**	**671, 50**	**0.87**	**0.15**	**13**
	DI	674, 50	649, 1	0.83	0.17	17
	mSR	600, 1	697, 1	0.75	0.22	25
	mNDI	451, 1	647, 1	0.83	0.17	17
	TVI	569, 1	771, 1	0.83	0.18	17
	SAVI	672, 50	649, 1	0.86	0.16	14
	NDVI	672, 50	649, 1	0.86	0.15	14

**Table 5 plants-14-03355-t005:** Summary of spectral channel parameters for VIs calculation obtained in this study for *Cucumis sativus L*. (first row) and *Lactuca sativa* L. (second row) and published datasets [53,54]. The index providing the highest *R*^2^ is highlighted in bold.

Formula	This Study (*Cucumis sativus L*. *Lactuca sativa* L.)				Database	
	*Chl*		*Car*			
	$\lambda_{1},\Delta \lambda_{1}$	$\lambda_{2},\Delta \lambda_{2}$	$\lambda_{1},\Delta \lambda_{1}$	$\lambda_{2},\Delta \lambda_{2}$	$\lambda_{1},\Delta \lambda_{1}$	$\lambda_{2},\Delta \lambda_{2}$
SR	454, 1	752, 1	**451, 1**	**748, 1**	640–760, -	780–1400, -
	650, 1	672, 50	**650, 1**	**671, 50**		
DI	703, 10	725, 1	732, 1	526, 1	678, -	500, -
	674, 50	649, 1	674, 50	649, 1		
mSR	751, 1	851, 1	451, 1	699, 1	705, -	750, -
	601, 1	696, 1	600, 1	697, 1		
**mNDI**	713, 50	692, 1	716, 50	644, 1	680, -	800, -
	**451, 1**	**649, 1**	451, 1	647, 1		
TVI	691, 50	715, 50	690, 50	714, 50	670, -	750, -
	597, 1	771, 1	569, 1	771, 1		
**SAVI**	**478, 50**	**747, 1**	746, 1	456, 1	665, 30	842, 115
	672, 50	650, 1	672, 50	649, 1		
NDVI	752, 1	453, 1	747, 1	451, 1	665, 30	865, 20
	672, 50	650, 1	672, 50	649, 1		

**Table 6 plants-14-03355-t006:** Plants description.

Species/Variety	Family	Maturity Stage	Leaf Morphology	Total Number of Leaf Samples
Lettuce (*Lactuca sativa* L.)/Credo	Asteraceae	30 days juvenile	Light green, mature, strongly wavy margin	50
Lettuce (*Lactuca sativa* L.)/Moskovskiy Parnikoviy	Asteraceae	30 days juvenile	Green, mature, oak-shaped and wavy with lobes	50
Cucumber (*Cucumis sativus*)/Mamlyuk F1	Cucurbitaceae	BBCH 10–19 4–5 true leaves	Green, young, palmately lobed and serrated	50
Cucumber (*Cucumis sativus*)/Mamlyuk F1	Solanaceae	BBCH 50–59 onset of flowering	Green, mature, palmately lobed and serrated	50

## Data Availability

The datasets generated during this study are available from the corresponding author upon reasonable request.

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
