# Peer review of "Variation in Assessment of Leaf Pigment Content from Vegetation Indices Caused by Positions and Widths of Spectral Channels"

_plants, 2025, doi:10.3390/plants14213355_

Round 1
Reviewer 1 Report
Comments and Suggestions for Authors
This manuscript investigates how variations in spectral channel positions and bandwidths and the choice of vegetation index (VI) formulas influence the accuracy of non-destructive pigment estimation. The authors measured hyperspectral reflectance and destructive pigment content in Lactuca sativa and Cucumis sativus leaves at different stages, simulated thousands of VI configurations, and evaluated prediction performance.
Strengths
Timely and relevant topic; quantifying the effect of channel configuration on VI accuracy is crucial for reproducibility (lines 17–35).
Dual species and multiple developmental stages strengthen generality (lines 135–151; 297–303).
Comprehensive simulation of both channel parameters and index formulas (lines 333–346).
Clear graphics showing sensitivity of R² and errors (Figures 4–7).
Limitations
Some sampling details (number of plants, leaf position, environmental conditions) are dispersed and incomplete (lines 281–303); they should be consolidated for replicability.
Discussion of underlying physiological mechanisms behind index performance differences is brief (lines 226–252).
The paper does not yet highlight commercialization potential (lines 253–264) or explicitly discuss transferability to other cultivars/species.
Specific Comments by Section
Abstract
Lines 17–21: Suggest adding that both Lactuca sativa and Cucumis sativus were used at contrasting developmental stages to underline the generality of findings.
Lines 24–27: The stated errors (42–77%) are striking; adding one phrase about how such errors can lead to suboptimal agronomic decisions (wrong fertilization, irrigation) would highlight applied relevance.
Introduction
Lines 42–51: Could also mention that low-cost RGB imaging and machine learning are emerging for on-farm monitoring. For instance, Tsaniklidis et al. (2025 Agronomy 15, 2294) shows how simple RGB images quantify plant organ size and greenness at low cost and are easily implemented commercially.
Lines 52–64 (Figure 1): Could hint at where errors may arise in each step (device development, ground truth data, calibration, routine use) to frame the subsequent analysis.
Lines 86–87: When noting that accuracy depends on calibration and environmental conditions, could also mention hydration status as a confounding factor. For example, Fanourakis et al. 2024 (Plant Growth Regulation 102, 485–496) demonstrates accurate retrieval of leaf moisture using full-spectrum CNN approaches.
Lines 125–133: Add one sentence that these findings can inform sensor standardization and the design of cost-effective devices for breeders and farmers.
Results
Lines 135–151 (Table 2): Pigment means and standard deviations given, but sample sizes per cultivar/stage are not reported. Clarifying n per group would strengthen confidence in the reported variability.
Lines 147–150: It is stated that heterogeneity in Cucumis sativus reflects developmental stages. Authors could briefly quantify how many plants per stage.
Lines 156–163 (Figure 3): Could annotate key absorption features in the figure to help readers interpret.
Lines 169–185 (Tables 3–4): Authors could discuss mechanistic reasons (leaf thickness, cuticle, internal scattering) instead of just reporting numbers.
Lines 190–198 (Figure 4): Suggest stating explicitly that this necessitates stable illumination and reference targets in practical settings.
Lines 200–206 (Figure 5): The trade-off between bandwidth and stability could be linked to commercial sensor design choices (e.g. multispectral cameras using 10–40 nm bands).
Lines 214–222 (Figure 6/7): The large variation in R² among index formulas is a key finding; could explicitly compare to published machine-learning models to show relative advantages.
Discussion
Lines 227–237 (3.1): Expand on physiological reasons: chlorophyll absorption slopes are steep; leaf internal structure differs between lettuce and cucumber; water content alters scattering. This links to hydration status.
Lines 244–252 (3.2): Discussion of calibration could stress that cultivar and developmental stage differences (as in lines 147–150; 299–303) produce distinct regression curves; therefore, universal calibrations are not feasible.
Lines 253–264 (3.3 Applicability): Currently emphasizes simplicity of VI-based concept. Add a paragraph on potential for low-cost implementation using RGB or narrowband sensors on handhelds, greenhouses or UAVs. Tsaniklidis et al. (2025) may be referred as an example of a commercializable approach.
Lines 253–264: Also note that leaf water status influences reflectance and may confound pigment estimation; integration of CNN models to retrieve moisture could enhance robustness and extend applicability to stress diagnostics.
Lines 265–279 (3.4 Standardization): Could also mention creating open spectral libraries including different species/stages to enable cross-platform calibration.
Materials and Methods
Lines 281–303 (4.1 Plants): Valuable information though scattered. Recommend consolidating: specify, for each species/stage, the number of plants, number of leaves per plant, leaf position (sun vs shade), and selection rationale. This ensures replicability.
Lines 287–295: Nutrient solutions described; could state if EC and pH were controlled and at what temperature/light conditions measurements occurred.
Lines 299–303: Add how many plants per stage and whether leaves were excised or measured in situ.
Lines 314–320 (4.3 Spectral data): Could add whether all measurements were made at the same time of day and under constant temperature/humidity to reduce variability.
Lines 315–320: You say three areas per leaf were measured; clarify how areas were chosen (mid-lamina, avoiding veins, constant distance from apex) and whether orientation relative to light source was fixed.
Lines 333–346 (4.4–4.5 Data processing and evaluation): Add a note on computation time per sample or total dataset to illustrate feasibility for high-throughput applications.
Conclusions
Consider adding: “These findings provide a basis for designing low-cost multispectral sensors and calibration protocols for reliable pigment estimation across species and developmental stages.”
Author Response
We thank Editor and Reviewers for insightful assessment and valuable comments. To address your remarks, we have revised the text. All changes are marked with green color. Please, find below our responses to all comments and questions included in the reviews.
Reviewer 1
Abstract
Comment 1: Lines 17–21: Suggest adding that both Lactuca sativa and Cucumis sativus were used at contrasting developmental stages to underline the generality of findings.
Response 1: We added a sentence about the measurement of the pigment content in leaves of Lactuca sativa L. and Cucumis sativus L. at contrasting developmental stages
Comment 2: Lines 24–27: The stated errors (42–77%) are striking; adding one phrase about how such errors can lead to suboptimal agronomic decisions (wrong fertilization, irrigation) would highlight applied relevance.
Response 2: We added a sentence (Lines 31-32) on how stated errors can lead to suboptimal agronomic decisions
Introduction
Comment 3: Lines 42–51: Could also mention that low-cost RGB imaging and machine learning are emerging for on-farm monitoring. For instance, Tsaniklidis et al. (2025 Agronomy 15, 2294) shows how simple RGB images quantify plant organ size and greenness at low cost and are easily implemented commercially.
Response 3: We added a phrase (Lines 50-53) about significance of low-cost implementations of spectral imager for on-farm monitoring.
Comment 4: Lines 52–64 (Figure 1): Could hint at where errors may arise in each step (device development, ground truth data, calibration, routine use) to frame the subsequent analysis.
Response 4: We added a sentence (Lines 67–70) about the sources of errors at different stages and clarified the scope to which the subsequent analysis is devoted
Comment 5: Lines 86–87: When noting that accuracy depends on calibration and environmental conditions, could also mention hydration status as a confounding factor. For example, Fanourakis et al. 2024 (Plant Growth Regulation 102, 485–496) demonstrates accurate retrieval of leaf moisture using full-spectrum CNN approaches.
Response 5: We added a sentence (Lines 67–70) stating that the plant’s condition in terms of non-pigment parameters could also serve as a confounding factor. We also mentioned the significance of monitoring hydration status and highlighted the potential of CNN-based approaches for this task in the later sections (Lines 203–206, Lines 326–327, and Lines 358–363).
Comment 6: Lines 125–133: Add one sentence that these findings can inform sensor standardization and the design of cost-effective devices for breeders and farmers.
Response 6: We added such a sentence (Lines 154–155).
Results
Comment 7: Lines 135–151 (Table 2): Pigment means and standard deviations given, but sample sizes per cultivar/stage are not reported. Clarifying n per group would strengthen confidence in the reported variability.
Response 7: We revised Table 2 (Line 174) to include the Total number of leaf samples for each cultivar and developmental stage. In addition, explanatory details were added in Lines 408–414 to specify the number of plants and leaves sampled per group.
Comment 8: Lines 147–150: It is stated that heterogeneity in Cucumis sativus reflects developmental stages. Authors could briefly quantify how many plants per stage.
Response 8: We revised Table 2 (Line 174) and Table 6 (Line 406) and added explanatory details in Lines 408–414 to specify the number of Cucumis sativus plants analyzed at each developmental stage.
Comment 9: Lines 156–163 (Figure 3): Could annotate key absorption features in the figure to help readers interpret.
Response 9: We added annotations of the key absorption features of plant pigments to Figure 3 (Line 190) and included a reference to the corresponding absorption spectra in Lines 178–181.
Comment 10: Lines 169–185 (Tables 3–4): Authors could discuss mechanistic reasons (leaf thickness, cuticle, internal scattering) instead of just reporting numbers.
Response 10: We added details (Lines 203–206) about the mechanistic reasons for the differences in the obtained results between the two species.
Comment 11: Lines 190–198 (Figure 4): Suggest stating explicitly that this necessitates stable illumination and reference targets in practical settings.
Response 11: We added details (Lines 264–268) about the requirements for the practical application of the obtained results.
Comment 12: Lines 200–206 (Figure 5): The trade-off between bandwidth and stability could be linked to commercial sensor design choices (e.g. multispectral cameras using 10–40 nm bands).
Response 12: We added a note (Lines 276–277) about the necessity of mentioning the trade-off between bandwidth and stability in commercial sensor design.
Comment 13: Lines 214–222 (Figure 6/7): The large variation in R² among index formulas is a key finding; could explicitly compare to published machine-learning models to show relative advantages.
Response 13: We added a description (lines 289–296) comparing VI-concept with published machine-learning approaches based on their resulting R² values.
Discussion
Comment 14: Lines 227–237 (3.1): Expand on physiological reasons: chlorophyll absorption slopes are steep; leaf internal structure differs between lettuce and cucumber; water content alters scattering. This links to hydration status.
Response 14: We expanded the description of the physiological reasons underlying the observed variation (Lines 325–329).
Comment 15: Lines 244–252 (3.2): Discussion of calibration could stress that cultivar and developmental stage differences (as in lines 147–150; 299–303) produce distinct regression curves; therefore, universal calibrations are not feasible.
Response 15: We added the corresponding note (Lines 337–339).
Comment 16: Lines 253–264 (3.3 Applicability): Currently emphasizes simplicity of VI-based concept. Add a paragraph on potential for low-cost implementation using RGB or narrowband sensors on handhelds, greenhouses or UAVs. Tsaniklidis et al. (2025) may be referred as an example of a commercializable approach.
Response 16: We added a paragraph dedicated to the potential applications of the obtained results for the design of low-cost devices (Lines 353–358).
Comment 17: Lines 253–264: Also note that leaf water status influences reflectance and may confound pigment estimation; integration of CNN models to retrieve moisture could enhance robustness and extend applicability to stress diagnostics.
Response 17: We emphasized the impact of leaf water status on plant reflectance and on strategies to overcome this limitation (Lines 358–363).
Comment 18: Lines 265–279 (3.4 Standardization): Could also mention creating open spectral libraries including different species/stages to enable cross-platform calibration.
Response 18: We added the corresponding note (Lines 369–371).
Materials and Methods
Comment 19: Lines 281–303 (4.1 Plants): Valuable information though scattered. Recommend consolidating: specify, for each species/stage, the number of plants, number of leaves per plant, leaf position (sun vs shade), and selection rationale. This ensures replicability.
Response 19: We revised Table 6 (Line 406) and added explanatory details in Lines 408–414 to clarify the experimental design.
Comment 20: Lines 287–295: Nutrient solutions described; could state if EC and pH were controlled and at what temperature/light conditions measurements occurred.
Response 20: We added information on the control of EC and pH values (Lines 394-395 and 402-404), as well as the environmental conditions during measurements (Lines 435-440).
Comment 21: Lines 299–303: Add how many plants per stage and whether leaves were excised or measured in situ.
Response 21: We clarified the number of plants used at each developmental stage (Lines 295-296 and 408-414) and specified that the spectral measurements were performed on excised leaves (Lines 415-418). The corresponding text has been revised accordingly.
Comment 22: Lines 314–320 (4.3 Spectral data): Could add whether all measurements were made at the same time of day and under constant temperature/humidity to reduce variability.
Response 22: We added a detailed description of the measurement conditions in Lines 435–440.
Comment 23: Lines 315–320: You say three areas per leaf were measured; clarify how areas were chosen (mid-lamina, avoiding veins, constant distance from apex) and whether orientation relative to light source was fixed.
Response 23: We clarified the procedure for selecting measurement areas and controlling leaf orientation in Lines 441–445.
Comment 24: Lines 333–346 (4.4–4.5 Data processing and evaluation): Add a note on computation time per sample or total dataset to illustrate feasibility for high-throughput applications.
Response 24: We added information on computation time ( Lines 468–473).
Conclusions
Comment 25: Consider adding: “These findings provide a basis for designing low-cost multispectral sensors and calibration protocols for reliable pigment estimation across species and developmental stages.”
Response 25: We added the corresponding note (Lines 496–499).

Reviewer 2 Report
Comments and Suggestions for Authors
The study touched on a quite interesting topic, the influence of wavelength and bandwidth on VI-based leaf pigment content prediction. I enjoyed reading the manuscript, but I believe it needs to be improved in several aspects before further evaluation.
The Introduction should provide a comprehensive literature review on this topic. Currently, it is unclear whether any existing studies have conducted similar research and what their findings were.
In Section 4, it is not clear exactly how many plants were sampled, how many times sampling was conducted, how many replicated measurements were recorded each time, and the total number of measurements taken in the study. Figure 7 is particularly concerning, as I hope the study included more than 19 samples.
I am also not sure if Table 2 makes sense. Lettuce and cucumber are two very high-level groups. Within these groups, the lettuce data involve different cultivars, and the cucumber data involve different developmental stages. Statistically, these data should not be grouped together, as they represent different “treatments” from an experimental design perspective. Only replicated measurements of the same plant cultivar at the same developmental stage should be grouped. I suggest redesigning Table 2 accordingly.
Line 167, the methodology needs further explanation. How were the optimal wavelength combinations selected?
Line 169, I disagree with this approach, as it could be misleading. In terms of optimal model performance, RMSE should always be prioritized over R². A high R² does not necessarily mean the model is accurate. The optimal spectral channel parameters in Tables 3 and 4 should therefore be determined based on RMSE.
Line 169, again the methodology needs further explanation. How were the optimal wavelength and band width combination selected?
Figures 4 and 5 should also focus on RMSE instead of R². Again, a high R² does not necessarily indicate high model accuracy.
The spectral wavelength range in Figures 4 and 5 is too wide to be meaningful. Under no scenario should the wavelength used for VI calculation vary by as much as (950–450) = 500 nm. It would be better to limit the range to maybe around 100 nm. The focus should be on how much the pigment prediction deteriorates when the wavelength is off by only a few nanometers, a more realistic and practical issue.
I think the authors should clearly cite the source studies for all the investigated VIs used in the study, specify what wavelengths were proposed in the original studies, and compare those to the optimal wavelengths identified in the current work.
Line 301, incomplete sentence.
Author Response
We thank Editor and Reviewers for insightful assessment and valuable comments. To address your remarks, we have revised the text. All changes are marked with green color. Please, find below our responses to all comments and questions included in the reviews.
Reviewer 2
Comment 1: The Introduction should provide a comprehensive literature review on this topic. Currently, it is unclear whether any existing studies have conducted similar research and what their findings were.
Response 1: We expanded Introduction with additional details on similar research and their findings (Lines 129–145).
Comment 2: In Section 4, it is not clear exactly how many plants were sampled, how many times sampling was conducted, how many replicated measurements were recorded each time, and the total number of measurements taken in the study. Figure 7 is particularly concerning, as I hope the study included more than 19 samples.
Response 2: We clarified the sampling design and measurement replication in Lines 408–414 and Table 6. Additionally, we clarified in Lines 295–296 that Figure 7 illustrates the results for 20 randomly selected samples from each dataset (each comprising 50 leaves per group).
Comment 3: I am also not sure if Table 2 makes sense. Lettuce and cucumber are two very high-level groups. Within these groups, the lettuce data involve different cultivars, and the cucumber data involve different developmental stages. Statistically, these data should not be grouped together, as they represent different “treatments” from an experimental design perspective. Only replicated measurements of the same plant cultivar at the same developmental stage should be grouped. I suggest redesigning Table 2 accordingly.
Response 3: We revised Table 2 (Line 174) to report the total number of leaf samples for each cultivar and developmental stage, rather than grouping data at the species level. In addition, explanatory details were added in Lines 408–414 to specify the number of plants and leaves sampled per group.
Comment 4: Line 167, the methodology needs further explanation. How were the optimal wavelength combinations selected?
Response 4: We clarified the methodology for selecting optimal wavelength combinations in Lines 192‑199.
Comment 5: Line 169, I disagree with this approach, as it could be misleading. In terms of optimal model performance, RMSE should always be prioritized over R². A high R² does not necessarily mean the model is accurate. The optimal spectral channel parameters in Tables 3 and 4 should therefore be determined based on RMSE.
Response 5: Indeed, RMSE is an important metric for assessing model accuracy. However, we believe that R² and RMSE should be evaluated together, as R² reflects the proportion of variance explained by the model, while RMSE represents the absolute prediction error. Combinations of these metrics ensures a more balanced assessment of model performance and prevents selecting indices that explain variance well but have large absolute errors. To address this issue, we updated Figures 4 and 5 and included RMSE maps alongside the existing R² maps. The indices reported in Tables 3 and 4 (Lines 214 and 215) were selected not only for maximum R² but also because they exhibited minimal RMSE within their respective groups, which is fully consistent with our findings.
Comment 6: Line 169, again the methodology needs further explanation. How were the optimal wavelength and band width combination selected?
Response 6: We clarified the methodology for selecting optimal wavelength combinations in Lines 192‑199.
Comment 7: Figures 4 and 5 should also focus on RMSE instead of R². Again, a high R² does not necessarily indicate high model accuracy.
Response 7: We updated Figures 4 and 5 and included RMSE maps alongside the existing R² maps.
Comment 8: The spectral wavelength range in Figures 4 and 5 is too wide to be meaningful. Under no scenario should the wavelength used for VI calculation vary by as much as (950–450) = 500 nm. It would be better to limit the range to maybe around 100 nm. The focus should be on how much the pigment prediction deteriorates when the wavelength is off by only a few nanometers, a more realistic and practical issue.
Response 8: We selected this spectral range to cover the range corresponding to widely spread hyperspectral and multispectral cameras. The clarification has been added (Lines 459–460). We agree with the reviewer’s comment and have addressed this point in Lines 251–254 by evaluating the effect of small deviations from the optimal central wavelengths.
Comment 9: I think the authors should clearly cite the source studies for all the investigated VIs used in the study, specify what wavelengths were proposed in the original studies, and compare those to the optimal wavelengths identified in the current work.
Response 9: We added Table 5 (Line 232) showing the wavelengths proposed in the original studies for all investigated VIs and the optimal wavelengths identified in the current research, and briefly compare these data (Lines 217–222).
Comment 10: Line 301, incomplete sentence.
Response 10: We revised this sentence.

Round 2
Reviewer 1 Report
Comments and Suggestions for Authors
The authors implemented every major and minor recommendation: Sampling and methodological clarity are now excellent. Mechanistic and physiological explanations were expanded. Applied/standardization aspects were added accurately.
Author Response
We thank Editor and Reviewers for insightful assessment and valuable comments!
Reviewer 2 Report
Comments and Suggestions for Authors
I do not believe the authors have addressed my comments sufficiently.
It remains unclear whether the added literature review in lines 130–146 is comprehensive. If not, additional relevant studies should be included.
In lines 139–146, studies [32–35] must be reviewed in detail. The differences between these studies and the current work, as well as their limitations, should be clearly highlighted.
The descriptions in lines 193–200 are still vague. When the authors state that they “systematically tested all combinations of spectral channel parameters,” how many combinations were tested in total? What were the wavelength range and step size, and the bandwidth range and step size, used in the testing? Was RMSE or R² used as the primary optimization target? The best RMSE and R² values could correspond to different parameter combinations.
Author Response
We thank Editor and Reviewers for insightful assessment and valuable comments. To address your remarks, we have revised the text. All changes are marked with green color. Please, find below our responses to all comments and questions included in the reviews.
Reviewer
Comment 1: It remains unclear whether the added literature review in lines 130–146 is comprehensive. If not, additional relevant studies should be included.
Response 1: We have added a few relevant references 32, 34, 35, 36, 37, 38.
Comment 2: In lines 139–146, studies [32–35] must be reviewed in detail. The differences between these studies and the current work, as well as their limitations, should be clearly highlighted.
Response 2: We have expanded Introduction in Lines 142-166 and added a few relevant references 40-52.
Comment 3: The descriptions in lines 193–200 are still vague. When the authors state that they “systematically tested all combinations of spectral channel parameters,” how many combinations were tested in total? What were the wavelength range and step size, and the bandwidth range and step size, used in the testing? Was RMSE or R² used as the primary optimization target? The best RMSE and R² values could correspond to different parameter combinations.
Response 3: We clarified this point in Lines 213-227, adding details on the total number of tested combinations, wavelength and bandwidth ranges, optimization criteria.
Round 3
Reviewer 2 Report
Comments and Suggestions for Authors
My comments have been addressed sufficiently.